# Proteomics and immunocharacterization of Asian mountain pit viper (*Ovophis monticola*) venom

Siravit Sitprija[1], Lawan Chanhome[2]*, Onrapak Reamtong[3], Tipparat Thiangtrongjit[3], Taksa Vasaruchapong[2], Orawan Khow[4], Jureeporn Noiphrom[4], Panithi Laoungbua[2], Arissara Tubtimyoy[1], Narongsak Chaiyabutr[2,4], Supeecha Kumkate[1]*

**1** Department of Biology, Faculty of Science, Mahidol University, Ratchathewi, Bangkok, Thailand, **2** Snake Farm, Queen Saovabha Memorial Institute, The Thai Red Cross Society, Bangkok, Thailand, **3** Department of Molecular Tropical Medicine and Genetics, Faculty of Tropical Medicine, Mahidol University, Ratchathewi, Bangkok, Thailand, **4** Department of Research and Development, Queen Saovabha Memorial Institute, The Thai Red Cross Society, Bangkok, Thailand

* lchanhome@yahoo.com (LC); supeecha.kum@mahidol.edu (SK)

**Data Availability Statement:** All relevant data are within the paper and its Supporting Information files.

## Abstract

The venomic profile of Asian mountain pit viper *Ovophis monticola* is clarified in the present study. Using mass spectrometry-based proteomics, 247 different proteins were identified in crude venom of *O. monticola* found in Thailand. The most abundant proteins were snake venom metalloproteases (SVMP) (36.8%), snake venom serine proteases (SVSP) (31.1%), and phospholipases $A_2$ (PLA$_2$) (12.1%). Less abundant proteins included L-amino acid oxidase (LAAO) (5.7%), venom nerve growth factor (3.6%), nucleic acid degrading enzymes (3.2%), C-type lectins (CTL) (1.6%), cysteine-rich secretory proteins (CRISP) (1.2%) and disintegrin (1.2%). The immunoreactivity of this viper's venom to a monovalent antivenom against green pit viper *Trimeresurus albolabris*, or to a polyvalent antivenom against hemotoxic venom was investigated by indirect ELISA and two-dimensional (2D) immunoblotting. Polyvalent antivenom showed substantially greater reactivity levels than monovalent antivenom. A titer for the monovalent antivenom was over 1:1.28x10$^7$ dilution while that of polyvalent antivenom was 1:5.12x10$^7$. Of a total of 89 spots comprising 173 proteins, 40 spots of predominantly SVMP, SVSP and PLA$_2$ were specific antigens for antivenoms. The 49 unrecognized spots containing 72 proteins were characterized as non-reactive proteins, and included certain types of CTLs and CRISPs. These neglected venom constituents could limit the effectiveness of antivenom-based therapy currently available for victims of pit viper envenomation.

## Introduction

Envenomation from snakebites affects over 2.7 million people in tropical and subtropical countries each year, leading to more than 130,000 deaths among victims [1]. Severe injuries and complications from bites also can lead to permanent disabilities and long-term health

**Funding:** This research was funded by the Center of Excellence on Biodiversity, grant number BDC-PG4-161009.

**Competing interests:** The authors have declared that no competing interests exist.

problems in survivors. In addition, the majority of snakebite victims have been reported within the productive age of the workforce [2]. This creates socio-ecomomic loss as seen in developing countries of Asia and Africa [3].

Currently, there are at least six known species of the venomous *Ovophis* genus (Family Viperidae, Subfamily Crotalinae) existed according to the phylogenetic and morphological analyses. Five closely related species are distributed across several geographical areas of the Asian mainland [4]. *Ovophis tonkinensis* occurs in northern Vietnam and southern China; *O. zayuensis* in southern China (Yunnan), northeastern India and Myanmar; *O. makazayazaya* in southern China (Sichuan, Yunnan), Taiwan and northern Vietnam; *O. convictus* is restricted to western Malaysia; and *O. monticola* is found in Nepal, northeastern India [5], southern China, Myanmar, southern Laos, central Vietnam and northern Thailand [4, 6]. The other member of this genus, *O. okinavensis*, inhabits Ryukyu Island of Japan [7].

In Thailand, the Asian mountain pit viper *O. monticola* is found in high-altitude mountains, particularly in the northern province of Chiang Mai [6]. It has a stout body with a short snout. Its triangular head is covered by small, smooth scales rather than large shields. These vipers also exhibit sexual dimorphism in body size, with an average male length of 49 cm and female length of 110 cm. These montane, terrestrial, nocturnal vipers generally live under the forest litter and prey on small mammals [8].

The medical significance of pit viper envenomation primarily relates to the hematotoxic activity on human victims. Severe clinical manifestation includes local damage (*e.g.*, painful oedema, tissue necrosis) and systemic injuries, including haemorrhage, coagulopathy and thrombocytopenia, critically resulting in high mortality and morbidity [5, 9].With advanced proteomic technology, the heterogeneity of snake venoms has progressively been elucidated. For the *Ovophis* spp., venomic profiles of *O. convictus* from western Malaysia, *O. tonkinensis* from northern Vietnam and southern China and Japanese hime habu *O. okinavensis* from Okinawa, Japan were recently reported. The abundance of four major enzymes namely snake venom serine proteinase (SVSP), phospholipases $A_2$ (PLA$_2$), L-amino acid oxidases (LAAO) and snake venom metalloproteases (SVMP) were dominant within all venoms. Among these enzymatic proteins, SVSP was found in the greatest proportion, accounting for 35–53% of all constituents. The second most abundant enzyme was PLA$_2$ ranging from 19–26%. In addition, various non-enzymatic proteins and peptides including cysteine-rich secretory proteins (CRISP), venom nerve growth factor (VNGF), venom endothelial growth factor (VEGF), kunitz peptides (KUN) and C-type lectins/snaclecs (CTL) were recorded, in varying amounts [10]. However, variation in snake venom composition occurs not only among distinct species but also among different population of the same species, due to ecological niches as well as availability of preys [11].

The present study aims to investigate the protein constituents of venom from the Asian mountain pit viper *O. monticola* found in Thailand. In addition, since there is no homospecific antivenom to *Ovophis* spp. venoms currently available, the therapeutic regime for bite victims depends largely on two types of antivenom: pit viper monovalent antivenom, raised against white-lipped green pit viper (*Trimeresurus albolabris*) venom; and polyvalent antivenom, produced against hematotoxic venom of *Calloselasma rhodostoma* (Malayan pit viper), *Daboia siamensis* (Russell's viper) and *T. albolabris*. Cross reactivity of *O. monticola* venom to these readily available antivenoms was therefore evaluated. Compositional profiles of immunoreactive versus non-reactive proteins in *O. monticola* venom were also clarified. Knowledge gained from this study not only extends the *Ovophis* spp. venomic database, but also can lead to better management and therapeutic approaches for mountain pit viper envenomation.

## Materials and methods

### Snakes, venom and antivenoms

All *O. monticola* pit vipers (Fig 1) were captured in the wild and transferred to Snake Farm, Queen Saovabha Memorial Institute (QSMI) before being quarantined. All procedures were performed following the safety protocol for working with venomous snakes (No. SN 001/2016). Routine snake care and the venom collection was conducted according to the specific protocol. All protocols were approved by the Ethic Committee of the Queen Saovabha Memorial Institute Animal Care and Use (No. QSMI-ACUC-02-2018) in accordance with the guideline of the National Research Council of Thailand. Information about individual snakes used in this study is shown in Table 1.

Monovalent antivenom against the green pit viper *T. albolabris* venom (batch no. TA00219; expiry date 08/10/2024) and hematotoxic polyvalent antivenoms (against the venom of *C. rhodostoma*, *D. siamensis* and *T. albolabris*) (batch no. HP 00118; expiry date 16/ 01/2023) produced by QSMI available as a freeze-dried F(ab')₂ form, isolated from horse immunoglobulins were used within their shelf-life. Following reconstitution, each milliliter of monovalent antivenom neutralized 0.7 mg of *T. albolabris* venom; one milliliter of hematotoxic polyvalent antivenom neutralized 0.7 mg of *T. albolabris* venom, 1.6 mg of *C. rhodostoma* venom and 0.6 mg of *D. siamensis* venom [12].

### O. monticola *venom preparation and one-dimensional sodium dodecyl sulfate polyacrylamide* gel electrophoresis (SDS-PAGE)

Crude venom of *O. monticola* was mixed with lysis buffer (containing of 1% Triton X-100 (Merck, Germany), 1% sodium dodecyl sulfate (SDS) (Merck, Germany), and 1% NaCl

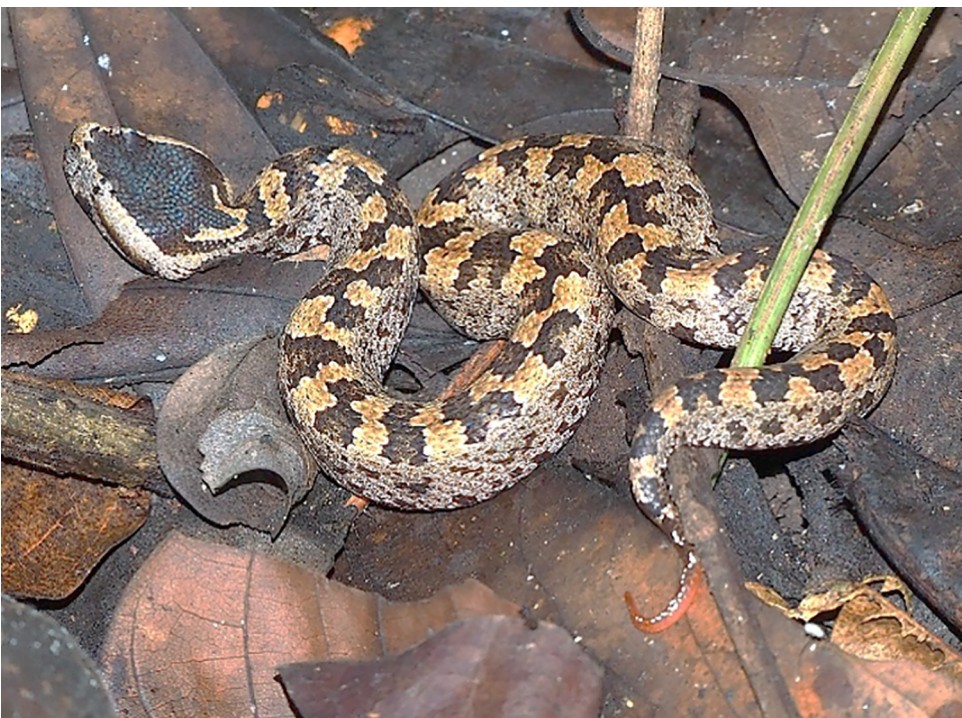

**Fig 1. A wild juvenile Asian mountain pit viper (O. monticola) found in Northern Thailand.** The venom of *O. monticola* was extracted and kept in individual 1.5 ml microcentrifuge tubes. After weighing, the fresh (liquid) venom was immediately frozen at -20˚C and lyophilized. The lyophilized venom was then pooled and stored at -20˚C until use.

**Table 1. Biological and geographical data for all snakes used in the study.**

| Species[a] | Voucher no.[b] | Sex[c] | Snout-Vent Length (cm) | Total Length (cm) | Locality[d] |
|---|---|---|---|---|---|
| *Ovophis monticola* | QSMI 1441 | F | 32.0 | 36.5 | Omkoi-Chiang Mai |
| *Ovophis monticola* | QSMI1443 | F | 32.0 | 36.5 | Doi Pui-Chiang Mai |
| *Ovophis monticola* | QSMI 1449 | M | 34.0 | 40.0 | Omkoi-Chiang Mai |
| *Ovophis monticola* | QSMI 1469 | M | 31.0 | 38.0 | Omkoi-Chiang Mai |
| *Ovophis monticola* | QSMI 1559 | M | 35.5 | 43.0 | Omkoi-Chiang Mai |

[a] The identification of *Ovophis monticola* was made by specialized veterinarians according to the identification key [8]. Key characters are body coloration and pattern: predominantly tan or reddish-grey with irregular short, black-edged crossbars or blotches along the vertebral ridge, including smaller irregular dark blotches on both sides of the body along the edges of the dorsal scales (Fig 1).

[b] Voucher no. was attached to each preserved snake after it died.

[c] Sex: F: Female; M: Male.

[d] District or subdistrict-province in Thailand where snakes were captured.

(Merck, Germany). The venom was estimated for protein concentration by Quick Start™ Bradford Protein Assay (Bio-Rad, USA). A 30 µg sample of *O. monticola* venom was separated by 12% sodium dodecyl sulfate polyacrylamide gel electrophoresis (SDS-PAGE) (Bio-Rad, USA) and stained by Coomassie R-250 solution (Bio-Rad, USA) as previously described [13]. The whole lane of venom was excised into 10 pieces and further subjected to in-gel digestion.

## Two-dimensional polyacrylamide gel electrophoresis (2DE)

A 100 µg protein was mixed with IPG sample buffer containing 8 M urea, 2% (w/v) 3-[(3-cholamidopropyl)dimethylammonio]-1-propanesulfonate (CHAPS), 15 mM dithiothreitol (DTT), and 0.5% IPG sample buffer [14]. Afterwards, the protein solution was rehydrated overnight into a non-linear immobilized pH gradient (IPG) strip (pH 3–10; Amersham Bioscience, USA). Isoelectric focusing (pI) was done using an Ettan IPGphorII instrument (Amersham Bioscience, USA) with the following settings: 30 V for 14 h, 200 V for 1 h, 500 V for 1 h, 1000 V for 1 h, 3500 V for 1 h, and 8000 V for 18 h. The IPG strips were equilibrated with DTT for 15 min and with iodoacetamine for 15 min. After incubation, the strips were placed onto a 12% SDS-PAGE gel. All three 2DE gels were stained with silver stain and the immunoreactive spots in these gels were excised and pooled for mass spectrometric analysis. Other two 2DE gels were used for immunoblotting.

## In-gel digestion

A mixture of 50% acetonitrile (ACN) in 50 mM ammonium bicarbonate was used for de-staining the blue color from gel slides [13]. Venom proteins were reduced by 4mM DTT and incubated at 60˚C for 15 min. The reduced proteins were further alkylated by 250 mM iodoacetamine (IAA) (Sigma-Aldrich, USA) and incubated at room temperature for 30 min in dark. The gel pieces were dehydrated by removing all solution and adding 100% ACN (Thermo Scientific, USA). For tryptic digestion, trypsin (Sigma-Aldrich, USA, T6567) in 50 mM ammonium bicarbonate (Sigma-Aldrich, USA) was added to rehydrate the gels, which were then incubated overnight at 37 ˚C. Peptide extraction was performed by adding 100% ACN and incubating for 15 min. The resulting solution was transferred into a new microcentrifuge tube and dried using a centrifugal concentrator (TOMY, Japan). The peptide mixtures were stored at -20˚C prior to mass spectrometric analysis.

## Mass spectrometric analysis

Venom peptides were dissolved in 0.1% formic acid (Sigma-Aldrich, USA) and subjected to an Ultimate® 3000 Nano-LC systems (Thermo Scientific, USA). The peptides were eluted and infused to a microTOF-Q II (Bruker, Germany). The acquisition was operated by HyStar™ version 3.2 (Bruker, Germany), and the resulting data were processed and converted to mascot generics format (.mgf) files using Compass DataAnalysis™ software version 3.4 (Bruker, Germany). A database search was performed using Mascot Daemon software (Matrix Science, USA) against the NCBI snake database with the following parameters: one missed cleavage site, variable modifications of carbamidomethyl (C) and oxidation (M), 0.8 Da for MS peptide tolerance and 0.8 Da for MS/MS tolerance. The significance threshold was set at 95%. Three biological replications were performed for protein identification.

## Indirect enzyme-linked immunosorbent assay (ELISA)

Immunoreactivity of protein antigens in *O. monticola* venom to monovalent and polyvalent antivenom was assessed by indirect enzyme-linked immunosorbent assay (ELISA) modified from Gawtham and colleagues [15]. Each well of a 96-well Maxisorp Nunc immune plate (Thermo Fisher Scientific, Denmark) was coated with 5 ng of *O. monticola* venom in 0.05 M carbonate/bicarbonate buffer pH 9.6 (50 μl/well) and kept at 4˚C overnight. Plates were washed three times with phosphate-buffered saline (PBS) pH 7.2, blocked by adding 200 μL of PBS containing 2% (*w/v*) bovine serum albumin (BSA) (Capricorn Scientific GmBH, Ebsdorfergrund, Germany) and incubated for 1.5 h at 37˚C. The plates were then washed three times with PBS-0.05% Tween (PBST). They were incubated again for 1 h at 37˚C with 50 μL of the serial dilution of either monovalent or polyvalent antivenom ($1:10^5$–$1:5\times10^7$ in 0.2% BSA-PBS). After washing the plate three times with PBST, 50 μL of horseradish peroxidase-conjugated goat anti-horse-IgG (Abcam, Cambridge, UK) in PBST (1:1000) was added into each well and further incubated for another hour at 37˚C. Plates were then washed three times with PBST. Fifty microliters of substrate solution (SureBlue TMB microwell peroxidase, Seracare Life Sciences, Milford, MA) was subsequently added to each well, and the plate was kept in the dark for 10 min at room temperature for the reaction to occur. The absorbance at 630 nm was read using a microplate reader (TECAN InfinitePro 200, Switzerland).

## Immunoblot analysis

The separated polypeptide spots from 2DE gels were transferred to nitrocellulose membrane for 90 min at 18 V on a Trans-blot semi-dry Transfer CellTM (Biorad) in semi-dry transfer buffer (48 mM Tris and 2.93 g glycine) pH 9.2 containing 20% methanol. The membranes were blocked using 5% (w/v) non-fat milk in PBS for 2 h at room temperature. The membranes were rinsed twice with PBS-T buffer pH 7.4 (8 mM sodium phosphate, 2 mM potassium phosphate, 140 mM NaCl, 2.7 mM KCl and 0.5% v/v Tween) for 30 s each. The blotted membranes were incubated with either monovalent or polyvalent antivenom (1:1000 in 0.2% BSA-PBS). After washing the membrane three times with PBS-T, 50 μL of horseradish peroxidase-conjugated goat anti-horse-IgG (Abcam, Cambridge, UK) in PBS-T (1:2000) was added, and the mixture was incubated for 1 h at ambient temperature under constant agitation. Membranes were washed three times with PBS-T buffer and one time with PBS. Immunogen spots were visualized by detection of peroxidase activity using Ultra TMB-Blotting Solution (ThermoFisher Scientific, UK).

### Statistical analysis

Quantitative data are presented as mean ± SEM. Statistical significance between groups was analyzed using standard t-tests or two-way ANOVA followed by the Bonferroni test. Significant p-values are indicated within the figure panels. Error bars indicate SEM.

## Results

### Proteomic analysis of *O. monticola* venom

Detectable proteins in venom of *O. monticola* were between 10–95 kDa (Fig 2A). Intense protein bands at 10, 15, 50 and 72 kDa and faint bands at 26, 28, 30, 34 and 95 kDa were recorded. There were 247 proteins found in *O. monticola* venom (a list of all proteins is shown in S1 Table). A classification of constituent proteins based on their biological properties is presented in Fig 2B. The most abundant proteins were snake venom metalloproteases (SVMP) (36.8%), snake venom serine proteases (SVSP) (31.1%), and phospholipases $A_2$ (PLA$_2$) (12.1%). Less abundant groups included L-amino acid oxidase (LAAO) (5.7%), venom nerve growth factor (3.6%), nucleic acid degrading enzymes (3.2%) C-type lectins (CTL) (1.6%), cysteine-rich secretory proteins (CRISP) (1.2%) and disintegrin (1.2%). Toxin biosynthesis and other proteins comprised 0.4%. The top 15 unique proteins identified in *O. monticola* venom are shown in Table 2.

### Immunoreactivity of protein antigens in *O. monticola* venom to monovalent and polyvalent antivenoms by indirect ELISA

Since there is no homospecific antivenom to *Ovophis* spp. venoms currently available, all pit viper envenoming victims are recommended to receive either monovalent antivenom (raised against *T. albolabris* venom) or hematotoxic polyvalent antivenom (produced against venoms of *C. rhodostoma*, *D. siamensis* and *T. albolabris*) to alleviate symptoms [16]. Indirect ELISA was used to determine the cross-reactivity of these antivenoms to *O. monticola* venom. Hematotoxic polyvalent antivenom exhibited a significantly greater level of immunoreactivity than the monovalent antivenom by 30–50% (up to the dilution 1: 1.6x10$^6$), $P<0.001$ (Fig 3). A titer for the monovalent antivenom was over 1:1.28x10$^7$ dilution, while that of hematotoxic polyvalent antivenom was 1:5.12x10$^7$ (Fig 3).

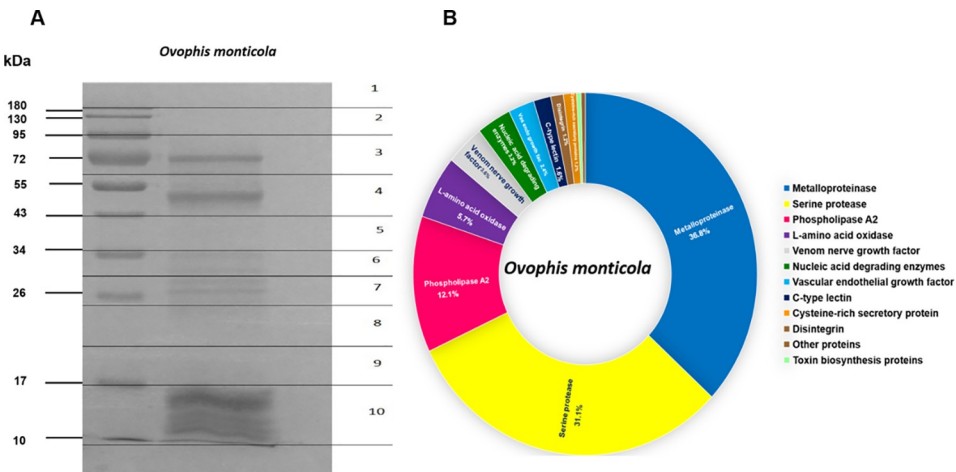

**Fig 2. Venomics of Asian mountain pit viper *O. monticola* from Thailand.** (A) Coomassie blue-stained 12% SDS-PAGE of *O. monticola* venom (30 μg) under reducing conditions. (B) Proteome classification of *O. monticola* venom; percentages indicate relative abundance (% of total venom proteins) of protein family in snake venom.

**Table 2. Fifteen most abundant unique proteins identified in *Ovophis monticola* venom.**

| No. | Accession no. | Protein | Score | MW[a] (Da) | No. of peptide | % Sequence coverage | pI[b] | emPAI[c] |
|---|---|---|---|---|---|---|---|---|
| 1 | sp\|P0C2D5.2\|OXLAPROFL | L-amino-acid oxidase | 534 | 3601 | 2 | 80 | 4.44 | 14.55 |
| 2 | sp\|O93517.1\|VM3S4GLOBR | Zinc metalloproteinase/disintegrin | 1293 | 11254 | 5 | 48.6 | 4.42 | 5.18 |
| 3 | sp\|Q9PRP4.1\|VSPFLACMR | Thrombin-like enzyme LMR-47 | 481 | 3168 | 2 | 100 | 4.31 | 3.82 |
| 4 | sp\|P0C590.1\|VSP2GLOUS | Thrombin-like enzyme calobin-2 | 481 | 2159 | 2 | 100 | 4.65 | 3.81 |
| 5 | sp\|C0HLA2.1\|VSP3LACMR | Thrombin-like enzyme LmrSP-3 | 335 | 2942 | 1 | 50 | 4.1 | 2.22 |
| 6 | sp\|P81478.1\|PA2A2TRIGA | Acidic phospholipase A2 2 | 621 | 13784 | 4 | 33.6 | 4.95 | 1.73 |
| 7 | sp\|C0HLA3.1\|VSP4LACMR | Snake venom serine protease LmrSP-4 | 1090 | 5841 | 2 | 62.3 | 4.28 | 1.60 |
| 8 | sp\|Q90W54.1\|OXLA_GLOBL | L-amino-acid oxidase | 2334 | 57056 | 19 | 32.7 | 6.52 | 1.37 |
| 9 | BAA01566.1 | Phospholipase A2 | 621 | 15697 | 3 | 40.6 | 4.99 | 1.35 |
| 10 | pdb\|1WVR\|A | Chain A, Triflin | 1099 | 24782 | 3 | 13.6 | 7.03 | 1.15 |
| 11 | sp\|Q7ZT99.1\|CRVPCROAT | Cysteine-rich venom protein catrin | 1099 | 26629 | 3 | 19.2 | 8.42 | 1.04 |
| 12 | sp\|Q7ZTA0.1\|CRVPAGKPI | Cysteine-rich venom protein piscivorin | 1099 | 26664 | 3 | 26.3 | 7.83 | 1.04 |
| 13 | sp\|E5L0E5.1\|VSPPAAGKPL | Venom plasminogen activator | 558 | 28060 | 5 | 12.8 | 5.78 | 1.02 |
| 14 | AAM80563.1 | Acidic phospholipase A2 | 372 | 15403 | 3 | 18.1 | 5.65 | 0.95 |
| 15 | sp\|P82896.1\|PA2A5TRIST | Acidic phospholipase A2 5 | 427 | 13870 | 2 | 32.8 | 4.72 | 0.93 |

[a]MW: Molecular weight (Dalton).

[b]pI: isoelectric point.

[c]emPAI: exponentially modified protein abundance index.

The information of identified proteins including NCBI accession number (Accession no.), protein name (Protein), protein score (Score), molecular weight of protein in Dalton unit (Da), Number of identified peptides (No. of peptide), % sequence coverage of the identified peptides (%Sequence coverage), isoelectric point of protein (pI) and exponentially modified protein abundance index for semi-quantification (emPAI) are demonstrated.

## *O. monticola* venom protein analysis by two-dimensional electrophoresis (2DE)

In order to explore the protein antigens present in *O. monticola* venom, crude venom was subjected to 2DE gel electrophoresis. There were 89 spots detected, with pI values ranging from 3 to 10 and MW from 10 to 95 kDa. Within the particular MW regions of 10–15, 30–34, 50 and 72–90 lies the greatest abundance of protein spots (Fig 4A). Using MALDI-TOF/TOF-MS/MS, all protein spots in 2DE gels were identified, and are listed in S2 Table. There were 461 different sequences, which correspond to 173 peptide accession identities.

## Immunoreactive proteins in *O. monticola* venom by immunoblot analysis

The immunoblot analysis was performed with either monovalent or polyvalent antivenom to characterize specific protein antigens within the *O. monticola* venom. Twenty-six immunoreactive spots were detected with monovalent antivenom, with pI values ranging from 3 to 6, and MW ranging from 17 to 95 kDa. Most of these spots were observed at MW 50 to 95 kDa and pI between 3 to 5 (Fig 4B). When probed with hematotoxic polyvalent antivenom, 40 immunoreactive spots were recorded with a broader range of pI values from 3 to 8, and MW ranging from 17 to 95 kDa. A high number of the spots were detected within a MW range of 40–55 kDa (Fig 4C). Comparing the immunoreactive spots obtained from polyvalent antivenom with all protein spots visualized by silver staining (Fig 4A), 49 spots (numbered 24–25, 31–32, 38–40, 42–45, 48–51, 54–73 and 76–89) were not immunologically recognized. These non-reactive spots were grouped according to their MW and pI values into three clusters. Cluster 1 appeared in the MW range from 26

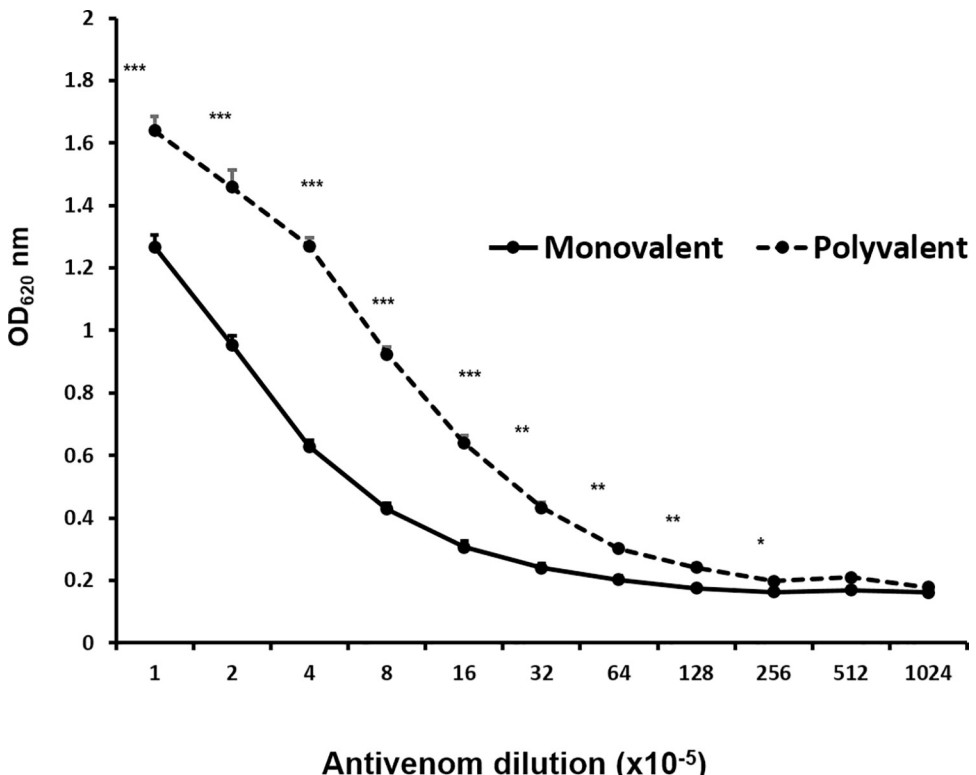

**Fig 3. Immunoreactivity of *O. monticola* venom to antivenoms.** Cross-reactivity of crude *O. monticola* venom to monovalent antivenom raised against green pit viper venom and polyvalent antivenom against snake hemotoxins. Data represent the mean ± SEM from two independent experiments; * $P < 0.05$, ** $P < 0.01$ and *** $P < 0.001$.

to 43 kDa, with high pI values (7–8); cluster 2 included those with MW ranging from 26 to 43 and with low pI values (3–5); and cluster 3 contained those with low MW from 10 to 26 kDa and with low pI values (3–5).

## Identification of immunoreactive and non-reactive peptides in *O. monticola* venom by LC-MS/MS

LC-MS/MS analysis revealed a total of 202 distinct sequences in *O. monticola* venom identified within 101 protein types that were immunologically reactive with a polyvenom. All

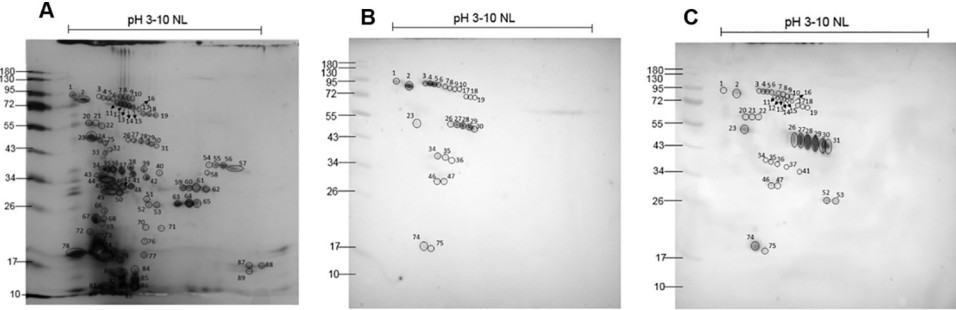

**Fig 4. The 2DE separations and immunoblot analysis of *O. monticola* venom.** (A) 2DE gels stained with silver stain; (B) 2D immunoblot of *O. monticola* proteins probed with monovalent antivenom and (C) polyvalent antivenom. Matched spots selected for subsequent LC-MS/MS analysis are marked and numbered.

immunoreactive proteins recognized by monovalent and polyvalent antivenom are listed in Table 3. However, from all 173 protein types appearing by silver staining, 72 proteins were left unrecognized by both antivenoms, and are shown in Table 4. Interestingly, the majority of these neglected peptides are well-known members of CTLs (e.g., C-type lectin, snaclec and galactose-binding lectin) and different CRISPs including okinavin, catrin and piscivorin.

## Discussion

The protein constituents within the venom of Asian mountain pit viper *O. monticola* from northern Thailand were investigated in the present study. SDS-PAGE revealed a protein band pattern ranging between MW 10–90 kDa with intense bands representing low MW proteins (10–15 kDa), those of 50 and 72 kDa, and faint bands between 30–50 kDa. This corresponded well to the spot pattern obtained from 2DE, where the clouds of protein spots were observed within MW regions of 10–15, 30–34, 50 and 72–90 kDa. The overall MW range of proteins in *O. monticola* venom was comparable to those from three other *Ovophis* species, namely *O. convictus*, *O. tonkinensis* and *O. okinavensis*. The pattern of dominant bands was most similar to the venom of O. *tonkinensis* found in China [10].

The proteomic profile showed that enzymatic components which are SVMP (36.8%), SVSP (31.1%), $PLA_2$ (12.1%) and LAAO (5.7%) mainly make up the venom of Asian mountain pit viper *O. monticola*. The overall composition of these major enzymes was comparable to those recently reported from venoms of *O. convictus*, *O. tonkinensis and O. okinavensis*, only content proportion seemed to be different. Among these three *Ovophis* spp., the most abundant proteins were of SVSP (35–53%), followed by $PLA_2$ (19–25%) and LAAO (5–17%). SVMP (11–19%) was detected at a lower percentage than in our venom [10]. Such venom variation in venom composition could would not only be attributed by speciation but also other factors including prey diversity reflecting different ecological habitats [11], snake sex, [17] and age [18, 19]. In addition, the variation in quantity of identified toxin types within snake venom might be resulted from the different quantitative approaches and accompanying calculation methods as well as proteomic database availability [20]. The 2DE indicated that *O. monticola* venom contained more acidic than basic protein spots. This finding was confirmed by our list of all identified peptides obtained from LC-MS/MS showing that the majority possessed pI values lower than 7. Our spot pattern also confirmed previous 2DE analysis of *Trimeresurus sumatranus* (another Viperid) venom, in which more proteins were identified in the acidic range than in Elapid venom [21]. The overall acidic properties of 4 main protein groups of vipers greatly contribute to the hemorrhagic and coagulopathic effect on victims [22, 23].

Our results revealed SVMPs as representing more than one-third of the entire *O. monticola* venom. The greatest amount found in *O. monticola* was sp|O93517.1|VM3S4_GLOBR Zinc metalloproteinase/disintegrin or disintegrin-like salmosin-4, first identified within Korean *Agkistrodon halys brevicaudus* snake venom [24]. SVMPs potentially inhibit platelet aggregation and integrin-dependent cell adhesion via interrupting glycoprotein IIb-IIIa/fibrinogen interaction and fibrinogenolysis [25, 26]. Additionally, SVMPs interact with the various types of cellular matrix and exerts the most haemorrhagic effect on hosts [27]. SVSPs were found to be second-most abundant in *O. monticola* venom. The majority are thrombin-like enzymes including sp|Q9PRP4.1|VSPF_LACMR thrombin-like enzyme LMR-47 and sp|Q9PRP4.1|VSPFLACMR thrombin-like enzyme calobin-2. Known as fibrinogen-clotting enzymes, they are common, and found in large amounts in the venoms of the genera *Agkistrodon*, *Bothrops*, *Lachesis* and *Trimeresurus* [28]. Thrombin-like enzymes demonstrate strong hydrolytic activity, primarily against triad residues of His57, Asp102 and Ser195 of fibrinogen [29]. Resembling thrombin, they act on blood plasma by forming friable and translucent clots which later

**Table 3. List of identified proteins in *Ovophis monticola* venom immunologically reactive with monovalent and polyvalent antivenoms.**

| Spot no. | Protein/peptide accession | Description [*Organisms*] | MW (Da) | Monovalent | Polyvalent |
|---|---|---|---|---|---|
| 1 | XP_029142019.1 | Zinc metalloproteinase-disintegrin-like atrolysin-A, partial [*Protobothrops mucrosquamatus*] | 60272 | √ | √ |
| | JAS04843.1 | Metalloproteinase type III 2b [*Crotalus horridus*] | 68297 | | |
| | JAS04684.1 | Metalloproteinase type III 1b [*Crotalus adamanteus*] | 67284 | | |
| | AAA03326.1 | Hemorrhagic toxin a (partial)[*Crotalus atrox*] | 46848 | | |
| | GBP06242.1 | Disintegrin and metalloproteinase domain-containing protein 12 [*Eumeta japonica*] | 199170 | | |
| 2 | XP_029142019.1 | Zinc metalloproteinase-disintegrin-like atrolysin-A, partial [*Protobothrops mucrosquamatus*] | 60272 | √ | √ |
| | JAS04843.1 | Metalloproteinase type III 2b [*Crotalus horridus*] | 68297 | | |
| | JAS04684.1 | Metalloproteinase type III 1b [*Crotalus adamanteus*] | 67284 | | |
| 3 | sp\|Q4VM07.1\|VM3VB_MACLB | Zinc metalloproteinase-disintegrin-like VLAIP-B (Snake venom metalloproteinase) | 68798 | √ | √ |
| | JAS04447.1 | Metalloproteinase type III 7 [*Agkistrodon piscivorus conanti*] | 68638 | | |
| | sp\|P0DM87.1\|VM2_TRIST | Zinc metalloproteinase-disintegrin stejnitin (Snake venom metalloproteinase) | 54401 | | |
| 4 | JAS04447.1 | Metalloproteinase type III 7 [*Agkistrodon piscivorus conanti*] | 68638 | √ | √ |
| | sp\|Q4VM07.1\|VM3VB_MACLB | Zinc metalloproteinase-disintegrin-like VLAIP-B | 68798 | | |
| 5 | sp\|Q4VM07.1\|VM3VB_MACLB | Zinc metalloproteinase-disintegrin-like VLAIP-B (Snake venom metalloproteinase) | 68798 | √ | √ |
| | sp\|P0DM87.1\|VM2_TRIST | Zinc metalloproteinase-disintegrin stejnitin (Snake venom metalloproteinase) | 54401 | | |
| | JAS04675.1 | Metalloproteinase type III 5 [*Crotalus adamanteus*] | 69463 | | |
| | JAS04447.1 | Metalloproteinase type III 7 [*Agkistrodon piscivorus conanti*] | 68638 | | |
| | XP_023086434.2 | disintegrin and metalloproteinase domain-containing protein 20-like [*Piliocolobus tephrosceles*] | 84212 | | |
| 6 | sp\|Q4VM07.1\|VM3VB_MACLB | Zinc metalloproteinase-disintegrin-like VLAIP-B (Snake venom metalloproteinase) | 68798 | √ | √ |
| | JAS04447.1 | Metalloproteinase type III 7 [*Agkistrodon piscivorus conanti*] | 68638 | | |
| | sp\|P0DM87.1\|VM2_TRIST | Zinc metalloproteinase-disintegrin stejnitin (Snake venom metalloproteinase) | 54401 | | |
| 7 | - | Not identified | | √ | √ |
| 8 | pdb\|1REO\|A | Chain A, Ahplaao | 55097 | √ | √ |
| | AAQ16182.1 | L-amino acid oxidase [*Trimeresurus stejnegeri*] | 58607 | | |
| 9 | pdb\|1REO\|A | Chain A, Ahplaao | 55097 | √ | √ |
| | sp\|A0A024BTN9.1\|OXLA_BOTSC | L-amino acid oxidase Bs29 | 56341 | | |
| 10 | sp\|A0A024BTN9.1\|OXLA_BOTSC | L-amino acid oxidase Bs29 | 56341 | √ | √ |
| 11 | pdb\|1REO\|A | Chain A, Ahplaao | 55097 | - | √ |
| | sp\|A0A024BTN9.1\|OXLA_BOTSC | L-amino acid oxidase Bs29 | 56341 | | |
| | sp\|P0C2D5.2\|OXLA_PROFL | L-amino-acid oxidase (Okinawa Habu apoxin protein-1) | 3601 | | |
| | sp\|P0C2D6.1\|OXLA_PROMU | L-amino-acid oxidase | 2929 | | |
| 12 | pdb\|1REO\|A | Chain A, Ahplaao | 55097 | - | √ |
| | sp\|A0A024BTN9.1\|OXLA_BOTSC | L-amino acid oxidase Bs29 | 56341 | | |
| | BAP39915.1 | L-amino acid oxidase [*Protobothrops elegans*] | 57339 | | |
| | sp\|P0DI84.1\|OXLA_VIPAA | L-amino-acid oxidase | 54714 | | |
| | sp\|C0HJE7.2\|OXLA_CRODU | L-amino acid oxidase bordonein-L | 58882 | | |
| | sp\|Q4F867.2\|OXLA_DABSI | L-amino-acid oxidase | 46343 | | |
| | sp\|X2JCV5.1\|OXLAA_CERCE | L-amino acid oxidase | 58520 | | |

(*Continued*)

**Table 3.** (Continued)

| Spot no. | Protein/peptide accession | Description [*Organisms*] | MW (Da) | Monovalent | Polyvalent |
|---|---|---|---|---|---|
| | sp\|A8QL51.1\|OXLA_BUNMU | L-amino-acid oxidase | 58774 | | |
| | sp\|P0C2D5.2\|OXLA_PROFL | L-amino-acid oxidase (Okinawa Habu apoxin protein-1) | 3601 | | |
| | sp\|A0A2U8QPE6.1\|OXLA_MICMP | L-amino acid oxidase | 57079 | | |
| | XP_026523888.1 | titin isoform X41 [*Notechis scutatus*] | 3637718 | | |
| 13 | pdb\|1REO\|A | Chain A, Ahplaao | 55097 | - | √ |
| | JAS04783.1 | L-amino acid oxidase 1b [*Crotalus horridus*] | 58587 | | |
| | sp\|P0DI84.1\|OXLA_VIPAA | L-amino-acid oxidase | 54714 | | |
| | BAP39915.1 | L-amino acid oxidase [*Protobothrops elegans*] | 57339 | | |
| | sp\|A0A024BTN9.1\|OXLA_BOTSC | L-amino acid oxidase Bs29 | 56341 | | |
| | sp\|C0HJE7.2\|OXLA_CRODU | L-amino acid oxidase bordonein-L | 58882 | | |
| | JAV01888.1 | BATXLAAO1 [*Bothrops atrox*] | 56625 | | |
| | sp\|Q4F867.2\|OXLA_DABSI | L-amino-acid oxidase | 46343 | | |
| | sp\|P0C2D5.2\|OXLA_PROFL | L-amino-acid oxidase (Okinawa Habu apoxin protein-1) | 3601 | | |
| | sp\|X2JCV5.1\|OXLAA_CERCE | L-amino acid oxidase | 58520 | | |
| | sp\|A0A2U8QPE6.1\|OXLA_MICMP | L-amino acid oxidase | 57079 | | |
| 14 | pdb\|1REO\|A | Chain A, Ahplaao | 55097 | - | √ |
| | sp\|A0A024BTN9.1\|OXLA_BOTSC | L-amino acid oxidase Bs29 | 56341 | | |
| | sp\|P0DI84.1\|OXLA_VIPAA | L-amino-acid oxidase | 54714 | | |
| | sp\|C0HJE7.2\|OXLA_CRODU | L-amino acid oxidase bordonein-L | 58882 | | |
| | sp\|P0C2D5.2\|OXLA_PROFL | L-amino-acid oxidase (Okinawa Habu apoxin protein-1) | 3601 | | |
| | JAV01888.1 | BATXLAAO1 [*Bothrops atrox*] | 56625 | | |
| | sp\|A0A2U8QPE6.1\|OXLA_MICMP | L-amino acid oxidase | 57079 | | |
| 15 | pdb\|1REO\|A | Chain A, Ahplaao | 55097 | - | √ |
| | sp\|A0A024BTN9.1\|OXLA_BOTSC | L-amino acid oxidase Bs29 | 56341 | | |
| | sp\|A0A2U8QPE6.1\|OXLA_MICMP | L-amino acid oxidase | 57079 | | |
| | XP_026523846.1 | Titin isoform X1 [*Notechis scutatus*] | 3675875 | | |
| 16 | pdb\|1REO\|A | Chain A, Ahplaao | 55097 | - | √ |
| | sp\|A0A024BTN9.1\|OXLA_BOTSC | L-amino acid oxidase Bs29 | 56341 | | |
| 17 | pdb\|1REO\|A | Chain A, Ahplaao | 55097 | √ | √ |
| | JAV01888.1 | BATXLAAO1 [*Bothrops atrox*] | 56625 | | |
| 18 | pdb\|1REO\|A | Chain A, Ahplaao | 55097 | √ | √ |
| | AAQ16182.1 | L-amino acid oxidase [*Trimeresurus stejnegeri*] | 58607 | | |
| | sp\|A0A024BTN9.1\|OXLA_BOTSC | L-amino acid oxidase Bs29 | 56341 | | |
| | sp\|P0DI84.1\|OXLA_VIPAA | L-amino-acid oxidase | 54714 | | |
| | JAV01888.1 | BATXLAAO1 [*Bothrops atrox*] | 56625 | | |
| | sp\|A0A2U8QPE6.1\|OXLA_MICMP | L-amino acid oxidase | 57079 | | |
| | sp\|A8QL51.1\|OXLA_BUNMU | L-amino-acid oxidase | 58774 | | |
| | sp\|P0C2D5.2\|OXLA_PROFL | L-amino-acid oxidase (Okinawa Habu apoxin protein-1) | 3601 | | |
| | XP_026523888.1 | Titin isoform X41 [*Notechis scutatus*] | 3637718 | | |

(*Continued*)

**Table 3.** (Continued)

| Spot no. | Protein/peptide accession | Description [*Organisms*] | MW (Da) | Monovalent | Polyvalent |
|---|---|---|---|---|---|
| 19 | sp\|A0A024BTN9.1\|OXLA_BOTSC | L-amino acid oxidase Bs29 | 56341 | √ | √ |
| 20 | BAN82126.1 | Serine protease, partial [*Ovophis okinavensis*] | 9035 | - | √ |
| | JAV51428.1 | Serine proteinase 12a [*Agkistrodon contortrix contortrix*] | 28885 | | |
| | XP_026529526.1 | Microtubule-actin cross-linking factor 1 isoform X1 [*Notechis scutatus*] | 838459 | | |
| 21 | sp\|P0C578.1\|VSP2_OVOOK | Thrombin-like enzyme okinaxobin-2 (Fibrinogen-clotting enzyme) | 2310 | - | √ |
| | JAV51428.1 | Serine proteinase 12a [*Agkistrodon contortrix contortrix*] | 28885 | | |
| | sp\|I2C090.1\|VCO3_OPHHA | Ophiophagus venom factor (Complement C3 homolog) | 183812 | | |
| | XP_026526061.1 | ALK and LTK ligand 1 [*Notechis scutatus*] | 21543 | | |
| | sp\|P85109.1\|VSP1_GLOBR | Thrombin-like enzyme kangshuanmei (Fibrinogen-clotting enzyme) | 26415 | | |
| | JAG68112.1 | Dynamin-binding protein [*Boiga irregularis*] | 90258 | | |
| 22 | JAV51428.1 | Serine proteinase 12a [*Agkistrodon contortrix contortrix*] | 28885 | - | √ |
| | BAN82126.1 | serine protease, partial [*Ovophis okinavensis*] | 9035 | | |
| | sp\|E5L0E5.1\|VSPPA_AGKPL | Venom plasminogen activator | 28060 | | |
| | sp\|Q5W958.1\|VSP20_BOTJA | Venom serine proteinase-like HS120 | 27797 | | |
| 23 | BAN82126.1 | Serine protease, partial [*Ovophis okinavensis*] | 9035 | √ | √ |
| | sp\|Q9PSN3.1\|VSP2_AGKBI | Thrombin-like enzyme bilineobin (Fibrinogen-clotting enzyme/Snake venom serine protease) | 26461 | | |
| | BAN82122.1 | Serine protease, partial [*Ovophis okinavensis*] | 8080 | | |
| | pdb\|2AIP\|A | Chain A, Protein C activator | 25090 | | |
| | sp\|C0HLA2.1\|VSP3_LACMR | Thrombin-like enzyme LmrSP-3 | 2942 | | |
| | ADI47563.1 | Serine protease, partial [*Echis ocellatus*] | 27233 | | |
| | sp\|P0C5B4.2\|VSPGL_GLOSH | Thrombin-like enzyme gloshedobin(Fibrinogen-clotting enzyme/Snake venom serine protease) | 28597 | | |
| | sp\|Q9DF66.1\|VSP3_PROJR | Snake venom serine protease 3 | 28007 | | |
| | pdb\|1OP0\|A | Chain A, Venom serine proteinase | 25318 | | |
| 26 | JAV51428.1 | Serine proteinase 12a [*Agkistrodon contortrix contortrix*] | 28885 | √ | √ |
| 27 | XP_029142018.1 | Zinc metalloproteinase-disintegrin jerdonitin [*Protobothrops mucrosquamatus*] | 58843 | √ | √ |
| | TSK34762.1 | Disintegrin and metalloproteinase domain-containing protein 12 [*Bagarius yarrelli*] | 146595 | | |
| | XP_032089254.1 | ras GTPase-activating-like protein IQGAP1 [*Thamnophis elegans*] | 189690 | | |
| 28 | XP_029142018.1 | Zinc metalloproteinase-disintegrin jerdonitin [*Protobothrops mucrosquamatus*] | 58843 | √ | √ |
| | sp\|P0DM87.1\|VM2_TRIST | Zinc metalloproteinase-disintegrin stejnitin | 54401 | | |
| | TSK34762.1 | Disintegrin and metalloproteinase domain-containing protein 12 [*Bagarius yarrelli*] | 146595 | | |
| 29 | XP_029142018.1 | Zinc metalloproteinase-disintegrin jerdonitin [*Protobothrops mucrosquamatus*] | 58843 | | |
| | sp\|P0DM87.1\|VM2_TRIST | Zinc metalloproteinase-disintegrin stejnitin (Snake venom metalloproteinase) | 54401 | √ | √ |
| | TSK34762.1 | Disintegrin and metalloproteinase domain-containing protein 12 [*Bagarius yarrelli*] | 215963 | | |
| | ETE65365.1 | putative helicase senataxin, partial [*Ophiophagus hannah*] | 146595 | | |
| 30 | XP_029142018.1 | Zinc metalloproteinase-disintegrin jerdonitin [*Protobothrops mucrosquamatus*] | 58843 | √ | √ |
| | sp\|P0DM87.1\|VM2_TRIST | Zinc metalloproteinase-disintegrin stejnitin (Snake venom metalloproteinase) | 54401 | | |
| 31 | XP_029142018.1 | Zinc metalloproteinase-disintegrin jerdonitin [*Protobothrops mucrosquamatus*] | 58843 | - | √ |
| 34 | JAS05371.1 | Serine proteinase 9d [*Sistrurus miliarius barbouri*] | 28266 | √ | √ |
| | sp\|P0DMH6.1\|VSP_BOTFO | Snake venom serine protease | 1729 | | |
| | sp\|Q8AY78.1\|VSP5M_TRIST | Snake venom serine protease 5 | 28117 | | |
| | sp\|Q8AY79.1\|VSPS2_TRIST | Beta-fibrinogenase stejnefibrase-2 (Snake venom serine protease) | 28010 | | |
| | sp\|Q5W958.1\|VSP20_BOTJA | Venom serine proteinase-like HS120 | 27797 | | |
| | sp\|Q71QH7.1\|VSPP_TRIST | Snake venom serine protease PA | 27933 | | |

(*Continued*)

**Table 3.** (Continued)

| Spot no. | Protein/peptide accession | Description [*Organisms*] | MW (Da) | Monovalent | Polyvalent |
|---|---|---|---|---|---|
| | XP_026540424.1 | Inositol hexakisphosphate and diphosphoinositol-pentakisphosphate kinase 1 isoform X1 [*Notechis scutatus*] | 135476 | | |
| 35 | sp\|P0DMH6.1\|VSP_BOTFO | Snake venom serine protease | 1729 | √ | √ |
| | sp\|E5L0E5.1\|VSPPA_AGKPL | Venom plasminogen activator | 28060 | | |
| | sp\|Q8AY78.1\|VSP5M_TRIST | Snake venom serine protease 5 | 28117 | | |
| | sp\|Q5W958.1\|VSP20_BOTJA | Venom serine proteinase-like HS120 | 27797 | | |
| | sp\|Q71QH7.1\|VSPP_TRIST | Snake venom serine protease PA | 27933 | | |
| | sp\|Q8AY79.1\|VSPS2_TRIST | Beta-fibrinogenase stejnefibrase-2 (Snake venom serine protease) | 28010 | | |
| 36 | JAS05372.1 | Serine proteinase 9c [*Sistrurus miliarius barbouri*] | 28221 | √ | √ |
| | JAS05371.1 | Serine proteinase 9d [*Sistrurus miliarius barbouri*] | 28266 | | |
| | JAV51414.1 | Serine proteinase 8 [*Agkistrodon contortrix contortrix*] | 28242 | | |
| | sp\|P0DMH6.1\|VSP_BOTFO | Snake venom serine protease | 1729 | | |
| | sp\|P0C5B4.2\|VSPGL_GLOSH | Thrombin-like enzyme gloshedobin (Fibrinogen-clotting enzyme/Snake venom serine protease) | 28597 | | |
| | ADI47574.1 | Serine protease, partial [*Echis coloratus*] | 28437 | | |
| | sp\|Q8AY78.1\|VSP5M_TRIST | Snake venom serine protease 5 | 28117 | | |
| | sp\|Q5W958.1\|VSP20_BOTJA | Venom serine proteinase-like HS120 | 27797 | | |
| | sp\|Q8AY79.1\|VSPS2_TRIST | Beta-fibrinogenase stejnefibrase-2 (Snake venom serine protease) | 28010 | | |
| | sp\|Q8UUJ2.2\|VSPUI_GLOUS | Snake venom serine protease ussurin; | 26184 | | |
| | sp\|Q71QH7.1\|VSPP_TRIST | Snake venom serine protease PA | 27933 | | |
| | XP_032092228.1 | Vitelline membrane outer layer protein 1 homolog isoform X1 [*Thamnophis elegans*] | 21236 | | |
| | JAI10638.1 | Vacuolar protein sorting-associated protein 18 homolog [*Crotalus adamanteus*] | 111967 | | |
| 37 | JAS05372.1 | Serine proteinase 9c [*Sistrurus miliarius barbouri*] | 28221 | - | √ |
| | JAS05371.1 | Serine proteinase 9d [*Sistrurus miliarius barbouri*] | 28266 | | |
| | sp\|P0DMH6.1\|VSP_BOTFO | Snake venom serine protease | 1729 | | |
| | JAV51414.1 | Serine proteinase 8 [*Agkistrodon contortrix contortrix*] | 28242 | | |
| | sp\|Q9PT41.1\|VSPF5_MACLB | Factor V activator (Lebetina viper venom FV activator/Snake venom serine protease | 28577 | | |
| | ADI47574.1 | Serine protease, partial [*Echis coloratus*] | 28437 | | |
| | sp\|Q8AY78.1\|VSP5M_TRIST | Snake venom serine protease 5; | 28117 | | |
| | XP_023086434.2 | Disintegrin and metalloproteinase domain-containing protein 20-like [*Piliocolobus tephrosceles*] | 84212 | | |
| | sp\|Q8AY79.1\|VSPS2_TRIST | Beta-fibrinogenase stejnefibrase-2 (Snake venom serine protease) | 28010 | | |
| 41 | - | Not identified | | - | √ |
| 46 | pdb\|1BQY\|A | Chain A, Plasminogen Activator | 25590 | √ | √ |
| | JAS04757.1 | Serine proteinase 1f [*Crotalus horridus*] | 28133 | | |
| | JAS04429.1 | Serine proteinase 13e [*Agkistrodon piscivorus conanti*] | 27985 | | |
| | JAS04417.1 | Serine proteinase 18b [*Agkistrodon piscivorus conanti*] | 27728 | | |
| | JAS04415.1 | Serine proteinase 19b [*Agkistrodon piscivorus conanti*] | 27782 | | |
| | sp\|Q072L7.1\|VSP_LACST | Snake venom serine protease | 27796 | | |
| | sp\|O13069.1\|VSP2_BOTJA | Thrombin-like enzyme KN-BJ 2 (Kinin-releasing and fibrinogen-clotting serine protease 2) | 26399 | | |
| | pdb\|4E7N\|A | Chain A, Snake-venom Thrombin-like Enzyme | 28333 | | |
| | XP_032089049.1 | Spectrin alpha chain, non-erythrocytic 1 [*Thamnophis elegans*] | 263010 | | |
| | XP_032064352.1 | Zinc finger protein 347-like [*Thamnophis elegans*] | 169270 | | |
| | sp\|Q9PT41.1\|VSPF5_MACLB | Factor V activator/Lebetina viper venom FV activatorSnake venom serine protease | 28577 | | |
| | ADI47574.1 | Serine protease, partial [*Echis coloratus*] | 28437 | | |
| 47 | JAS04757.1 | Serine proteinase 1f [*Crotalus horridus*] | 28133 | √ | √ |

(*Continued*)

**Table 3.** (Continued)

| Spot no. | Protein/peptide accession | Description [*Organisms*] | MW (Da) | Monovalent | Polyvalent |
|---|---|---|---|---|---|
| | pdb\|1BQY\|A | Chain A, Plasminogen Activator | 25590 | | |
| | JAS04417.1 | Serine proteinase 18b [*Agkistrodon piscivorus conanti*] | 27728 | | |
| | JAS04429.1 | Serine proteinase 13e [*Agkistrodon piscivorus conanti*] | 27985 | | |
| | JAS04415.1 | Serine proteinase 19b [*Agkistrodon piscivorus conanti*] | 27782 | | |
| | JAV01826.1 | BATXSVSP10 [*Bothrops atrox*] | 28606 | | |
| | pdb\|4E7N\|A | Chain A, Snake-venom Thrombin-like Enzyme | 26370 | | |
| | sp\|Q6T5L0.2\|VSPSH_GLOSH | Alpha-fibrinogenase shedaoenase (Snake venom serine protease) | 26399 | | |
| | sp\|O13069.1\|VSP2_BOTJA | Thrombin-like enzyme KN-BJ 2 (Kinin-releasing and fibrinogen-clotting serine protease 2) | 27876 | | |
| | sp\|Q71QI0.1\|VSP07_TRIST | Snake venom serine protease homolog KN7 | 28703 | | |
| | XP_015671556.1 | Snake venom serine protease [*Protobothrops mucrosquamatus*] | 28023 | | |
| | JAS04671.1 | Serine proteinase 3b [*Crotalus adamanteus*] | 28890 | | |
| | QHR82809.1 | Serine protease 2 [*Vipera anatolica senliki*] | 28084 | | |
| | sp\|A8QL53.1\|VSP1_NAJAT | Snake venom serine protease NaSP | 31117 | | |
| | XP_026523831.1 | Integrin alpha-4 [*Notechis scutatus*] | 114850 | | |
| 52 | JAS05359.1 | Cysteine-rich secretory protein 1c [*Sistrurus tergeminus*] | 26787 | - | √ |
| 53 | ETE67131.1 | Keratin, type II cytoskeletal 1, partial [*Ophiophagus hannah*] | 240496 | - | √ |
| 74 | sp\|A8E2V8.1\|PA2A_TRIGS | Acidic phospholipase A2 Tgc-E6 | 15678 | √ | √ |
| | sp\|P0DJJ7.1\|PA2A_OVOMO | Acidic phospholipase A2 Omo-E6 | 3261 | | |
| | JAV51451.1 | Phospholipase A2 1a [*Agkistrodon contortrix contortrix*] | 15952 | | |
| | sp\|Q6EAN6.1\|PA2A_SISTE | Acidic phospholipase A2 homolog sistruxin APrecursor | 15419 | | |
| | XP_032088152.1 | Group IIE secretory phospholipase A2-like [*Thamnophis elegans*] | 17310 | | |
| | sp\|Q7ZTA6.1\|PA2AB_CROVV | Acidic phospholipase A2 Cvv-E6b | 15429 | | |
| | AAB28455.1 | Phospholipase A2 isozyme III, PLA2-III [*Trimeresurus gramineus*] | 13716 | | |
| | JAV01879.1 | BATXPLA5 [*Bothrops atrox*] | 15504 | | |
| | sp\|P06860.1\|PA2BX_PROFL | Basic phospholipase A2 PL-X | 13971 | | |
| | AAB28454.1 | Phospholipase A2 isozyme IV, PLA2-IV [1] [*Trimeresurus gramineus*] | 13783 | | |
| | sp\|C0HJC1.1\|PA2_BOTLA | Acidic phospholipase A2 BlatPLA2 | 13881 | | |
| 75 | sp\|A8E2V8.1\|PA2A_TRIGS | Acidic phospholipase A2 Tgc-E6 | 15678 | √ | √ |
| | sp\|P0DJJ7.1\|PA2A_OVOMO | Acidic phospholipase A2 Omo-E6 | 3261 | | |
| | pdb\|1C1J\|A | Chain A, Basic phospholipase A2 | 13888 | | |
| | JAS04499.1 | Phospholipase A2 1s [*Agkistrodon piscivorus conanti*] | 15776 | | |
| | sp\|P82896.1\|PA2A5_TRIST | Acidic phospholipase A2 5 | 13870 | | |
| | sp\|D6MKR0.1\|PA2A6_CROHD | Acidic phospholipase A2 CH-E6 | 15498 | | |
| | sp\|Q7ZTA6.1\|PA2AB_CROVV | Acidic phospholipase A2 Cvv-E6b | 15429 | | |
| | JAV51451.1 | Phospholipase A2 1a [*Agkistrodon contortrix contortrix*] | 15952 | | |
| | sp\|P86907.1\|PA2A_BOTAM | Acidic phospholipase A2 | 13858 | | |
| | sp\|C9DPL5.1\|PA2A1_BOTPI | Acidic phospholipase A2 BpirPLA2-I | 13627 | | |
| | sp\|C0HLF0.1\|PA2_POROP | Basic phospholipase A2 | 14042 | | |
| | sp\|C0HJC1.1\|PA2_BOTLA | Acidic phospholipase A2 BlatPLA2 | 13881 | | |
| | sp\|P86456.1\|PA2A4_BOTAL | Acidic phospholipase A2 SpII RP4 | 13733 | | |
| | QHR82796.1 | Phospholipase A2 3 [*Vipera anatolica senliki*] | 17437 | | |

**Table 4. List of non-immunoreactive proteins/peptides in *Ovophis monticola* venom.**

| | Protein/peptide accession | Description [*Organisms*] | MW (Da) | Spot no. |
|---|---|---|---|---|
| 1 | AAZ75628.1 | Kallikrein-Phi4, partial [*Philodryas olfersii*] | 26827 | 45, 49, 50 |
| 2 | BAN82001.1 | Galactose binding lectin, partial [*Protobothrops flavoviridis*] | 17654 | 80, 82, 84, 85 |
| 3 | BAN82034.1 | Serine protease, partial [*Protobothrops flavoviridis*] | 22377 | 55 |
| 4 | BAN82147.1 | Cysteine rich secretory protein [*Ovophis okinavensis*] | 26920 | 59, 60, 61, 62 |
| 5 | BAN82148.1 | Galactose binding lectin [*Ovophis okinavensis*] | 18480 | 80, 82 |
| 6 | BAN82149.1 | C-type lectin alpha subunit [*Ovophis okinavensis*] | 17686 | 87, 88 |
| 7 | ETE59238.1 | Fascin-3, partial [*Ophiophagus hannah*] | 14941 | 48 |
| 8 | ETE60526.1 | Trichohyalin, partial [*Ophiophagus hannah*] | 80507 | 51 |
| 9 | ETE61374.1 | Dynein heavy chain 8, axonemal [*Ophiophagus hannah*] | 284552 | 72 |
| 10 | ETE64295.1 | Glycerol-3-phosphate acyltransferase 4 [*Ophiophagus hannah*] | 50772 | 86 |
| 11 | ETE66458.1 | Helicase SRCAP, partial [*Ophiophagus hannah*] | 494261 | 87 |
| 12 | ETE70787.1 | N6-adenosine-methyltransferase 70 kDa subunit, partial [*Ophiophagus hannah*] | 59797 | 89 |
| 13 | JAI12774.1 | Leucine-rich repeat-containing protein 7-like [*Crotalus adamanteus*] | 163679 | 45 |
| 14 | JAS04407.1 | Serine proteinase 6 [*Agkistrodon piscivorus conanti*] | 28115 | 43,44, 48, 76 |
| 15 | JAS04411.1 | Serine proteinase 2 [*Agkistrodon piscivorus conanti*] | 28333 | 49, 50, 55, 57, |
| 16 | JAS04568.1 | Phospholipase A2 1b [*Boiga irregularis*] | 16906 | 67, 72, 73, 78 |
| 17 | JAS04670.1 | Serine proteinase 3c [*Crotalus adamanteus*] | 28849 | 45, 49, 50, 54, 55, 56, 57, 58 |
| 18 | JAS04734.1 | Cysteine-rich secretory protein [*Crotalus adamanteus*] | 26612 | 59, 60, 61, 62 |
| 19 | JAS04742.1 | Serine proteinase 9d [*Crotalus horridus*] | 28299 | 49 |
| 20 | JAS04748.1 | Serine proteinase 6 [*Crotalus horridus*] | 28594 | 54, 55, 56, 57 |
| 21 | JAS05249.1 | Serine proteinase 2 [*Sistrurus tergeminus*] | 28326 | 49 |
| 22 | JAS05472.1 | C-type lectin 2 [*Sistrurus miliarius barbouri*] | 18147 | 87 |
| 23 | JAS05484.1 | Cysteine-rich secretory protein 1b [*Sistrurus miliarius barbouri*] | 26772 | 61 |
| 24 | JAV51425.1 | Serine proteinase 15a [*Agkistrodon contortrix contortrix*] | 28940 | 54, 55, 56, 57 |
| 25 | JAV51455.1 | C-type lectin 9a [*Agkistrodon contortrix contortrix*] | 18657 | 80 |
| 26 | pdb|1BK9|A | Chain A, Phospholipase A2 | 13964 | 72 |
| 27 | pdb|1GMZ|A | Chain A, Phospholipase A2 | 13850 | 67 |
| 28 | pdb|1JZN|A | Chain A, Galactose-specific lectin | 16281 | 80, 82 |
| 29 | pdb|1WVR|A | Chain A, Triflin | 24782 | 59, 60, 61, 62 |
| 30 | pdb|3JR8|A | Chain A, Phospholipase A2 bothropstoxin-2 | 13985 | 67 |
| 31 | sp|A0A1I9KNP0.1|VSPH1_VIPAA | Vaa serine proteinase homolog 1 | 28909 | 49 |
| 32 | sp|A8QL56.1|VSP1_OPHHA | Alpha- and beta-fibrinogenase OhS1 | 28637 | 45, 49, 50 |
| 33 | sp|B0VXW0.1|OXLA_SISCA | L-amino-acid oxidase | 58532 | 77 |
| 34 | sp|B0ZT25.1|VSPH_PROJR | Snake venom serine protease homolog | 28776 | 58 |
| 35 | sp|C0HLA1.1|VSP2_LACMR | Thrombin-like enzyme LmrSP-2 (Snake venom serine protease) | 3271 | 25, 32 |
| 36 | sp|E5AJX2.1|VSP_VIPBN | Snake venom serine protease nikobin | 28197 | 45, 49 |
| 37 | sp|J3S832.1|VSPB_CROAD | Snake venom serine proteinase 11 | 28033 | 58 |
| 38 | sp|J3S833.1|VSP2_CROAD | Snake venom serine proteinase 2 | 28298 | 49 |
| 39 | sp|K4LLQ2.1|VSP_BOTBA | Thrombin-like enzyme barnettobin (Snake venom serine protease) | 27567 | 38, 39 |
| 40 | sp|O13057.1|VSP2_PROFL | Snake venom serine protease 2 | 28623 | 56 |
| 41 | sp|O93421.2|VSPPE_GLOHA | Snake venom serine protease pallase | 26031 | 49 |
| 42 | sp|P0DJG8.1|CRVP_HELAG | Helicopsin | 2618 | 61 |
| 43 | sp|P0DL18.1|CRVP_OVOOK | Cysteine-rich venom protein okinavin | 3496 | 59, 60, 61 |
| 44 | sp|P0DM36.1|LECG_AGKPI | C-type lectin APL | 16195 | 80, 82, 84, 85 |
| 45 | sp|P81114.1|SLA4_TRIAB | Snaclec alboaggregin-A subunit beta | 14357 | 81, 83, 86 |
| 46 | sp|P81176.1|VSP1_GLOBL | Thrombin-like enzyme halystase (Snake venom serine protease) | 26466 | 49 |

*(Continued)*

**Table 4.** (Continued)

| | Protein/peptide accession | Description [*Organisms*] | MW (Da) | Spot no. |
|---|---|---|---|---|
| 47 | sp\|P82981.1\|VSP2_AGKCO | Thrombin-like enzyme contortrixobin/Fibrinogen-clotting enzyme (Snake venom serine protease) | 25396 | 54, 55, 56, 57, 58 |
| 48 | sp\|Q27J47.1\|VSPPA_LACMU | Venom plasminogen activator LV-PA | 28044 | 44 |
| 49 | sp\|Q71QJ4.1\|VSP04_TRIST | Snake venom serine protease homolog KN4 | 28685 | 49, 50, 55, 56, 57 |
| 50 | sp\|Q7SZE2.1\|VSPD_GLOUS | Bradykinin-releasing enzyme KR-E-1 (Snake venom serine protease) | 25335 | 49 |
| 51 | sp\|Q7T229.1\|VSPH_BOTJR | Snake venom serine protease homolog | 28636 | 54, 55, 56, 57 |
| 52 | sp\|Q7ZT99.1\|CRVP_CROAT | Cysteine-rich venom protein catrin | 26629 | 59 |
| 53 | sp\|Q7ZTA0.1\|CRVP_AGKPI | Cysteine-rich venom protein piscivorin | 26664 | 60, 61, 62 |
| 54 | sp\|Q8AY81.1\|VSPST_TRIST | Thrombin-like enzyme stejnobin (Fibrinogen-clotting enzyme/Snake venom serine protease) | 29309 | 25, 32 |
| 55 | sp\|Q91053.1\|VSP1_GLOUS | Thrombin-like enzyme calobin-1 (Snake venom serine protease) | 28889 | 49 |
| 56 | sp\|Q9YGJ2.1\|VSP1_GLOHA | Snake venom serine protease pallabin | 28662 | 49 |
| 57 | XP_015671564.1 | Snake venom serine protease serpentokallikrein-1 [*Protobothrops mucrosquamatus*] | 88822 | 56 |
| 58 | XP_023418723.1 | Disintegrin and metalloproteinase domain-containing protein 17 [*Cavia porcellus*] | 92703 | 64 |
| 59 | XP_024069019.3 | Disintegrin and metalloproteinase domain-containing protein 17 [*Terrapene carolina triunguis*] | 99769 | 64 |
| 60 | XP_025414344.1 | Disintegrin and metalloproteinase domain-containing protein 9 [*Sipha flava*] | 138126 | 54 |
| 61 | XP_026527653.1 | Laminin subunit alpha-1 [*Notechis scutatus*] | 331363 | 80 |
| 62 | XP_026535629.1 | Dynein heavy chain 8, axonemal [*Notechis scutatus*] | 511281 | 66 |
| 63 | XP_026540213.1 | Regulatory solute carrier protein family 1 member 1 [*Notechis scutatus*] | 37705 | 80 |
| 64 | XP_026541175.1 | N-acetylated-alpha-linked acidic dipeptidase-like protein [*Notechis scutatus*] | 81911 | 50 |
| 65 | XP_028906446.1 | Disintegrin and metalloproteinase domain-containing protein 17 [*Ornithorhynchus anatinus*] | 94942 | 63 |
| 66 | XP_032078796.1 | Laminin subunit alpha-1 [*Thamnophis elegans*] | 339446 | 85 |
| 67 | XP_032080246.1 | Centromere-associated protein E [*Thamnophis elegans*] | 308791 | 59 |
| 68 | XP_032085798.1 | 60S ribosomal protein L6 isoform X1 [*Thamnophis elegans*] | 30312 | 42, 81 |
| 69 | XP_032088226.1 | Forkhead-associated domain-containing protein 1 [*Thamnophis elegans*] | 137802 | 81 |
| 70 | XP_032091805.1 | Glial fibrillary acidic protein [*Thamnophis elegans*] | 52228 | 88 |
| 71 | XP_039181676.1 | Snake venom serine protease-like isoform X1 [*Crotalus tigris*] | 25811 | 49 |
| 72 | XP_039181680.1 | Snake venom serine proteinase 12-like [*Crotalus tigris*] | 24539 | 56 |

are degraded due to non-functional cross-linked structures, leading to coagulopathy and hypo-fibrinogenemia [30, 31]. However, in terms of enzyme proportion, our findings contrast with previous reports on the transcriptomic analysis of *O. okinavensis* venom glands, which mainly contained SVSP (93.1%) and relatively little SVMP (4.2%) [32]. It remains unclear whether the microenvironment within the venom gland might preferentially activate or interfere with the functioning of newly synthesized enzymes.

Compared with SVMP and SVSP, we detected smaller amounts of $PLA_2$ and LAAO in *O. monticola* venom. $PLA_2$ hydrolyzes phospholipids at the sn-2 position, generating fatty acids and lysophospholipids [33]. Group II $PLA_2$ is expressed exclusively in the venoms of the Viperidae [34]. We found both acidic and basic $PLA_2$ subtypes in *O. monticola* venom. It is noteworthy that the acidic $PLA_2$ (sp\|P81478.1\|PA2A2TRIGA and sp\|P82896.1\|PA2A5TRIST were present in high quantities partially contributing to more acidic properties of the venom. These were previously found to trigger oedema [35]. $PLA_2$ elicits inflammatory responses through the overproduction of pro-inflammatory cytokines (such as TNFα, IL-1β and IL-6) largely by immunocompetent cells (monocytes, neutrophils and mast cells) [36].

LAAOs can act in concert with PLA$_2$ in local inflammatory reactions. LAAOs from *Calloselasma rhodosthoma* venom were shown to induce superoxide anion and hydrogen peroxide production by human neutrophils [37]. The most abundant LAAO in *O. monticola* venom is sp|P0C2D5.2|OXLA_PROFL, also known as Okinawa habu apoxin protein-1. This protein was first characterized from the venom of *Protobothrops flavoridis* to induce apoptosis in glioma cells [38]. The roles of both PLA$_2$ and LAAOs in snake venoms are multi-faceted. Their catalytic as well as cytotoxic properties have been extensively investigated for pharmaceutical potential against cancers and other diseases [39].

Due to the unavailability of homospecific antivenom to *Ovophis* spp. venoms, all pit viper envenoming victims usually receive either monovalent antivenom (raised against *T. albolabris* venom) or hematotoxic polyvalent antivenom (produced against venoms of *C. rhodostoma*, *D. siamensis* and *T. albolabris*) [16]. The latter gave considerably higher immunoreactive levels (30–50%) to *O. monticola* venom proteins than the former. Relatively greater levels of reactivity of the polyvalent antivenom was previously reported with the venoms of *C. rhodostoma*, *Hypnale hypnale* and *Trimeresurus hageni*, and even *Trimeresurus albolabris* when compared with those of monovalent antivenom [40]. With the combination of 2DE immunoblotting and LC-MS/MS analyses, we found that hematotoxic polyvalent antivenom reacted with a wider range of proteins and peptides accounting for 58% of the entire range of proteins and covering all major enzymatic groups. Nonetheless, we were able to observe that an array of LAAOs and SVSPs did not react with a monovalent antivenom specific only to *T. albolabris* venom. This finding suggests the shared antigenic epitopes particularly from *Ovophis*, *Calloselasma* and *Trimeresurus* venoms used to generate antivenom. In this context, proteomic analysis of Malayan pit viper *C. rhodostoma* venom revealed a similar SVMP dominance (41.17%), with other major constituents of snaclec (26.3%) and SVSP (14.9%) [41]. A study of phylogenetic relationships based on geographic distribution and mitochondrial and nuclear gene sequences also demonstrated that *O. monticola* is less distantly separated from *C. rhodostoma* than from *T. albolabris* [42]. Thus, the antivenom against immunogenic epitopes from *C. rhodostoma* venom should be further investigated for the possible adjunctive treatment of *O. monticola* bite victims.

Our current study revealed that 72 proteins (42% of venom proteins) were left unrecognized by both antivenoms. The majority of immunologically non-reactive proteins have low molecular mass. They include a number of SVSPs, PLA$_2$ and certain SVMP. The poor immunogenicity of these low molecular venomic proteins has been obviously reported, although some possess high toxicity [43, 44]. In addition, an array of non-enzymatic CTLs such as galactose-binding lectins, snaclec, alboaggregin A, and CRISPs such as triflin, okinavin, catrin, and piscivotin were found unrecognizable by antivenom. This reflects the difference in antigenic abundance between *O. monticola* venom and those venoms employed to generate horse immunoglobulins, as the proportions of CTLs and CRISPs in our *O. monticola* venom were only 1.6% and 1.2%, respectively. Nonetheless, their biological impacts on host cells and tissues should not be neglected. In the context of CTLs, alboaggregin A was shown to bind strongly with platelet glycoproteins IB and VI, and hence, activated platelet aggregation [45]. In addition, evidence of enhanced platelet activation and thrombotic microangiopathy-like symptoms has been documented with other related snaclecs [46]. In terms of CRISPs, the unique pdb| 1WVR|A chain A triflin, as well as sp|P0DL18.1|CRVP_OVOOK okinavin from the related hime habu *O. okinavensis* were not reactive with the antivenoms. They have been previously described to have a calcium channel-impairing effect, leading to aberrations in muscle contraction [47, 48]. Considering the pathophysiological effects, our findings address the suite of protein targets which could be additional antigens for future antivenomic design. Furthermore, in order to alleviate the symptoms of mountain pit viper envenomation, these would

facilitate the development of specific drug schemes allowing patients to recover more quickly. The proteomic profile of *O. monticola* venom not only provides insight into the venomic phenotypes reflecting the evolutionary path among Viperid snakes, but also accelerates the discovery of novel candidates for medical and pharmaceutical use.

## Supporting information

**S1 Raw image.**
(PDF)

**S1 Table. List of all proteins found in *Ovophis monticola* venom.**
(XLSX)

**S2 Table. List of proteins in 89 spots in *Ovophis monticola* venom.** Crude venom was subjected to 2DE gel. Proteins were separated in the first dimension in the pH range 3–10.
(DOCX)

## Acknowledgments

We thank David Anderson for manuscript proofreading and the Central Equipment Unit, Faculty of Tropical Medicine, Mahidol University, Thailand for proteomics facilities.

## Author Contributions

**Conceptualization:** Siravit Sitprija, Lawan Chanhome, Onrapak Reamtong, Taksa Vasaruchapong, Orawan Khow, Supeecha Kumkate.

**Data curation:** Lawan Chanhome, Onrapak Reamtong.

**Formal analysis:** Lawan Chanhome, Onrapak Reamtong, Supeecha Kumkate.

**Investigation:** Onrapak Reamtong, Tipparat Thiangtrongjit, Orawan Khow, Jureeporn Noiphrom, Supeecha Kumkate.

**Methodology:** Lawan Chanhome, Onrapak Reamtong, Tipparat Thiangtrongjit, Orawan Khow, Jureeporn Noiphrom, Arissara Tubtimyoy, Supeecha Kumkate.

**Project administration:** Siravit Sitprija.

**Resources:** Lawan Chanhome, Taksa Vasaruchapong, Panithi Laoungbua.

**Software:** Onrapak Reamtong, Arissara Tubtimyoy.

**Supervision:** Siravit Sitprija, Lawan Chanhome, Narongsak Chaiyabutr, Supeecha Kumkate.

**Validation:** Onrapak Reamtong, Supeecha Kumkate.

**Writing – original draft:** Onrapak Reamtong, Supeecha Kumkate.

**Writing – review & editing:** Siravit Sitprija, Lawan Chanhome, Supeecha Kumkate.

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
