## [Decision Letter · Decision Letter 0]

13 Jul 2021

PONE-D-21-18686

Proteomics and immunocharacterization of Asian Mountain pit viper (Ovophis  monticola) venoms

PLOS ONE

Dear Dr. Kumkate,

Thank you for submitting your manuscript to PLOS ONE. After careful consideration, we feel that it has merit but does not fully meet PLOS ONE’s publication criteria as it currently stands. Therefore, we invite you to submit a revised version of the manuscript that addresses the points raised during the review process.

We look forward to receiving your revised manuscript.

Kind regards,

Rafael Ximenes

Academic Editor

PLOS ONE

Journal Requirements:

2. In your Methods section, please provide additional location information, including geographic coordinates for the data set if available.

4. We understand that you extracted venom from Ovophis monticola for this study. In your Methods section, please provide additional details regarding the source of the animal (Asian Mountain pit viper). Please provide the geographic coordinates and as well as any further details about the animal (e.g., sex, description of appearance) to ensure reproducibility of the analyses.

[No]. 

7. PLOS ONE now requires that authors provide the original uncropped and unadjusted images underlying all blot or gel results reported in a submission’s figures or Supporting Information files. This policy and the journal’s other requirements for blot/gel reporting and figure preparation are described in detail at https://journals.plos.org/plosone/s/figures#loc-blot-and-gel-reporting-requirements and https://journals.plos.org/plosone/s/figures#loc-preparing-figures-from-image-files. When you submit your revised manuscript, please ensure that your figures adhere fully to these guidelines and provide the original underlying images for all blot or gel data reported in your submission. See the following link for instructions on providing the original image data: https://journals.plos.org/plosone/s/figures#loc-original-images-for-blots-and-gels. 

8.  We note that Figure 1 in your submission contain copyrighted images. All PLOS content is published under the Creative Commons Attribution License (CC BY 4.0), which means that the manuscript, images, and Supporting Information files will be freely available online, and any third party is permitted to access, download, copy, distribute, and use these materials in any way, even commercially, with proper attribution. For more information, see our copyright guidelines: http://journals.plos.org/plosone/s/licenses-and-copyright.

a) You may seek permission from the original copyright holder of Figure 1 to publish the content specifically under the CC BY 4.0 license. 

Reviewers' comments:

Reviewer's Responses to Questions

**Comments to the Author**

1. Is the manuscript technically sound, and do the data support the conclusions?

Reviewer #1: Yes

Reviewer #2: Yes

2. Has the statistical analysis been performed appropriately and rigorously? 

Reviewer #1: N/A

Reviewer #2: N/A

3. Have the authors made all data underlying the findings in their manuscript fully available?

Reviewer #1: Yes

Reviewer #2: Yes

4. Is the manuscript presented in an intelligible fashion and written in standard English?

Reviewer #1: Yes

Reviewer #2: Yes

5. Review Comments to the Author

Reviewer #1: Referee comments:

In this work, the authors demonstrate by venomics approaches, a comprehensive characterization of the venom proteome and immunocharacterization of Asian Mountain pit viper (Ovophis monticola) venoms, a medically important snake from Thailand, found at high altitude areas of several mountain ranges of East, South and Southeast Asia. Moreover the authors analyzed the immunoreactivity of monospecific (monovalent) and polyspecific (polyvalent) antivenoms against green pit viper Trimeresurus albolabris and hemotoxic venoms, respectively. The authors present a study with satisfactory quality in methodology and data generation. However the manuscript has an inventory of toxins and immunoreactive proteins without coherence with the scientific literature and poor consistency with some cited references. Furthermore, in many topics of the manuscript the authors present a lot of information without any reference. For example, the authors don't compare the results regarding 2D immunoblotting with any previously study. I suggest that this manuscript should be reword to improve the quality about the data presentation. I'm sending some notes

- Abstract

Line 33  ..... venoms was investigated by indirect ELISA..... Line 37.....were subjected to 2D immunoblotting

These related methodologies would be more appropriate if they were introduced together.

- Introduction

. Line 46-50 - The geographical distribution data deserve references. The same for the sentences introducing biological data about pit viper from lines 51-58.... length of 110 cm.

Lines 59: Incidence of pit vipers  bites, including those from mountain pit vipers, have been documented in their known ranges???.

The lack of references should be revised in all manuscript...

-Materials and methods

Line 92 : Snake venom,

- Are there more information about the venom's sample used in this study? Number of specimens? Localities? size/age...

Line 121: ...a non-linear immobilized pH gradient (IPG) strip (pH 3-10 Amersham Bioscience, USA)....

- Why the authors choose the non-linear gradient instead of linear gradient, since the propose were excised immunoreactive spots for mass spectrometric analysis? Just for curious

- Lack references in almost all methods. The references for 2D immunoblotting is required

Results

Line 208: Change Ovophis / Ovophis sp or genera Ovophis

- Are there any published information about neutralize activity of monovalent and polyvalent antivenoms against O. monticola venom? This information should be provided or I suggest that an experiment to determine the lethal dose 50% should be done.

Table 2. List of identified proteins ... immunologically reacted ...

- If the MW of the spots content reacted/ non-reacted proteins could be provided, this information can improve the quality table.

Discussion

The Authors should revise the references from the first paragraph.... For example: the statement about hematotoxic potential of SVMP, SVSP and PLA2 is already a scientific consensus. Therefore, there are many sources specially from the seminal studies that corroborate with this question. However, the Authors present references from venom proteomic analysis of other snakes to support this question. On the other hand, the authors present coherency in the literature proposed as reference to address the mechanism of action of P-III metalloproteinase. I suggest that his paragraph must be reworded.

Line 208: The presence of class P-II metalloproteinase, which possesses metalloproteinase and disintegrin-like domians was also observed.????

Line 303: The combination of SVMP and SVSP observed in O. monticola venom is responsible for prey attack, incapacitation, and digestion.???

Reviewer #2: It is an interesting work to be published in PLOS. This article describes describe proteomics and immuno characterization of snake Ovophis monticola venom. I have the following comments that the authors may like to consider:

Line 95

- Provide the total number of individuals, age, sex, locality (adults or young people, males, females)

- It is important to provide some functional analysis

Discussion:

Despite little published literature on Ovophis monticola:

Please, include in the discussion this reference:

Pandey, D. P., Chaudhary, B., & Ram Shrestha, B. (2021). Documentation of a proven Mountain Pitviper (Ovophis monticola) envenomation in Kathmandu, Nepal, with its distribution ranges: implications for prevention and control of pitviper bites in Asia. Journal of venom research, 11, 1–6.

Mainly about clinical characteristic of poisoning by this specie and lack of specific treatment.

Provide conclusion or future perspectives in the last paragraphy about immunocharacterization results/ envenomation treatment.

Line 527

Exclude the number 75.

6. PLOS authors have the option to publish the peer review history of their article (what does this mean?). If published, this will include your full peer review and any attached files.

Reviewer #1: No

Reviewer #2: No

---

## [Author Response · Author response to Decision Letter 0]

20 Aug 2021

Responses to reviewers

Reviewer #1: 

In this work, the authors demonstrate by venomics approaches, a comprehensive characterization of the venom proteome and immunocharacterization of Asian Mountain pit viper (Ovophis monticola) venoms, a medically important snake from Thailand, found at high altitude areas of several mountain ranges of East, South and Southeast Asia. Moreover the authors analyzed the immunoreactivity of monospecific (monovalent) and polyspecific (polyvalent) antivenoms against green pit viper Trimeresurus albolabris and hemotoxic venoms, respectively. The authors present a study with satisfactory quality in methodology and data generation. However the manuscript has an inventory of toxins and immunoreactive proteins without coherence with the scientific literature and poor consistency with some cited references. Furthermore, in many topics of the manuscript the authors present a lot of information without any reference. For example, the authors don't compare the results regarding 2D immunoblotting with any previously study. I suggest that this manuscript should be reword to improve the quality about the data presentation.

Response: We thank the reviewer for suggestion. We rewrote introduction and discussion to improve data presentation. Relevant references were cited in every part to enhance literature coherence. Biological and geographical details regarding all snakes were also provided in materials and methods (Table 1; page 5, line 105). 

- Abstract

Line 33 ..... venoms was investigated by indirect ELISA..... Line 37.....were subjected to 2D immunoblotting

These related methodologies would be more appropriate if they were introduced together.

Response: We edited an abstract according to reviewer’s suggestion. 

- Introduction

. Line 46-50 - The geographical distribution data deserve references. The same for the sentences introducing biological data about pit viper from lines 51-58.... length of 110 cm.

Lines 59: Incidence of pit vipers bites, including those from mountain pit vipers, have been documented in their known ranges???.

The lack of references should be revised in all manuscript...

Response : We rewrote introduction with relevant references cited in a revised version of manuscript; ref no.6 and 8).

-Materials and methods

Line 92 : Snake venom,

- Are there more information about the venom's sample used in this study? Number of specimens? Localities? size/age...

Response: Biological and geographical information regarding all snakes is provided as a table in materials and methods (Table 1; page 5 line 105). 

Line 121: ...a non-linear immobilized pH gradient (IPG) strip (pH 3-10 Amersham Bioscience, USA)....

- Why the authors choose the non-linear gradient instead of linear gradient, since the propose were excised immunoreactive spots for mass spectrometric analysis? Just for curious

Response: According to our previous experiments, we found that the non-linear immobilized pH gradient (IPG) could provide the well separation of protein in venoms. Therefore, we chose the NL strip for this research.

- Lack references in almost all methods. The references for 2D immunoblotting is required.

Response: In a revised materials and methods of manuscripts, we provided relevant references for all methods used. For 2DE immunoblotting, a reference is doi: 10.3390/toxins6051526 as reference page 7, line 137 (ref no.14) 

Results

Line 208: Change Ovophis / Ovophis sp or genera Ovophis

- Are there any published information about neutralize activity of monovalent and polyvalent antivenoms against O. monticola venom? This information should be provided or I suggest that an experiment to determine the lethal dose 50% should be done.

Table 2. List of identified proteins ... immunologically reacted ...

- If the MW of the spots content reacted/ non-reacted proteins could be provided, this information can improve the quality table.

Response: We added a column of MW of proteins in a revised Table 2. In this particular study, we focused mainly on providing the inventory of all protein constituents of Asian mountain pit viper O. monticola venom. Those with antigenic properties reacted with monovalent and hematotoxic polyvalent antivenom were given. However, we did not perform neutralizing experiments.

Discussion

The Authors should revise the references from the first paragraph.... For example: the statement about hematotoxic potential of SVMP, SVSP and PLA2 is already a scientific consensus. Therefore, there are many sources specially from the seminal studies that corroborate with this question. However, the Authors present references from venom proteomic analysis of other snakes to support this question. On the other hand, the authors present coherency in the literature proposed as reference to address the mechanism of action of P-III metalloproteinase. I suggest that his paragraph must be reworded.

Line 208: The presence of class P-II metalloproteinase, which possesses metalloproteinase and disintegrin-like domians was also observed.????

Line 303: The combination of SVMP and SVSP observed in O. monticola venom is responsible for prey attack, incapacitation, and digestion.???

Response: We thank the reviewer for comments on discussion. We rewrote discussion by updating and improving literature coherence. 

Reviewer #2: 

It is an interesting work to be published in PLOS. This article describes describe proteomics and immuno characterization of snake Ovophis monticola venom. I have the following comments that the authors may like to consider:

Line 95

- Provide the total number of individuals, age, sex, locality (adults or young people, males, females)

- It is important to provide some functional analysis

Response: Biological and geographical information regarding all snakes is provided as a table in materials and methods (Table 1; page 5 line 105). In this particular study, we focused mainly on providing the inventory of all protein constituents of Asian mountain pit viper O. monticola venom. Those with antigenic properties reacted with monovalent and hematotoxic polyvalent antivenom were given. However, we did not perform the experiments to analyse the functions. 

Discussion:

Despite little published literature on Ovophis monticola:

Please, include in the discussion this reference:

Pandey, D. P., Chaudhary, B., & Ram Shrestha, B. (2021). Documentation of a proven Mountain Pitviper (Ovophis monticola) envenomation in Kathmandu, Nepal, with its distribution ranges: implications for prevention and control of pitviper bites in Asia. Journal of venom research, 11, 1–6.

Mainly about clinical characteristic of poisoning by this specie and lack of specific treatment.

Provide conclusion or future perspectives in the last paragraphy about immunocharacterization results/ envenomation treatment.

Line 527

Exclude the number 75.

Response: We cited the above literature as a reference in a revised version of manuscript (ref no.5).

---

## [Decision Letter · Decision Letter 1]

6 Oct 2021

PONE-D-21-18686R1Proteomics and immunocharacterization of Asian Mountain pit viper (Ovophis  monticola) venomsPLOS ONE

Dear Dr. Kumkate,

Thank you for submitting your manuscript to PLOS ONE. After careful consideration, we feel that it has merit but does not fully meet PLOS ONE’s publication criteria as it currently stands. Therefore, we invite you to submit a revised version of the manuscript that addresses the points raised during the review process.

We look forward to receiving your revised manuscript.

Kind regards,

Rafael Ximenes

Academic Editor

PLOS ONE

Journal Requirements:

Additional Editor Comments (if provided):

Based on the new round of peer-round some minor concerns have arisen. I recommend a minor revision before acceptance of the manuscript.

Reviewers' comments:

Reviewer's Responses to Questions

**Comments to the Author**

1. If the authors have adequately addressed your comments raised in a previous round of review and you feel that this manuscript is now acceptable for publication, you may indicate that here to bypass the “Comments to the Author” section, enter your conflict of interest statement in the “Confidential to Editor” section, and submit your "Accept" recommendation.

Reviewer #1: All comments have been addressed

Reviewer #3: All comments have been addressed

2. Is the manuscript technically sound, and do the data support the conclusions?

Reviewer #1: Yes

Reviewer #3: Yes

3. Has the statistical analysis been performed appropriately and rigorously? 

Reviewer #1: Yes

Reviewer #3: Yes

4. Have the authors made all data underlying the findings in their manuscript fully available?

Reviewer #1: Yes

Reviewer #3: Yes

5. Is the manuscript presented in an intelligible fashion and written in standard English?

Reviewer #1: No

Reviewer #3: Yes

6. Review Comments to the Author

Reviewer #1: The authors rewrote the manuscript and provide relevant references and endeavored to comply with all referee requests.I just emphasize that only Ovophis don't belong to a taxonomic system. This task dont were revide for authors. However. The authors present a study with high quality in methodology, data generation and data analysis, producing significant contribution for toxinology field. Thus, this work is suitable for publication on PLOS ONE.

Reviewer #3: The article highlights the spectrometry-based proteomics and the immunoreactivity of the venom of the Ovophis monticola viper. It is a work based on an excellent premise and the methodologies are very well defined. I recommend the work for publication. However, some minor revisions are needed.

1. In the session "Two-dimensional polyacrylamide gel electrophoresis (2DE)" (Material and Methods), the authors state that a 100 µg sample of venom was used. However, since the protein concentrations were measured using the Lowry method, please review whether these 100 µg are venom or protein.

2. In the session “Proteomic analysis of O. monticola venom” (Results), the first paragraph deals with a methodological aspect.

3. The same observation applies to the following topics. Instead of repeating the concepts of the methodology, the objective could be presented if evaluating that result, since this information is not present in the text.

4. In “Immunoreactivity of protein antigens in O. monticola venom to monovalent and polyvalent antivenoms by indirect ELISA”, the information “Since there is no homospecific antivenom to Ovophis venoms currently available, all pit viper envenoming victims usually receive either monovalent antivenom (raised against T. albolabris venom) or hematotoxic polyvalent antivenom (produced against venoms of C. rhodostoma, D. siamensis and T. albolabris) to alleviate symptoms” is applicable to the discussion, requiring references.

5. Figures and tables should be self-explanatory. In the case of table 2, a legend stating the meaning of MW, PI, emPI must be provided.

6. Discussion: What is the toxinological consequence related to the fact that the venom has more acidic than basic protein spots? How does this interfere with symptoms related to accident? Is this positive or negative regarding the immunoreactivity of the venom, especially with the polyvalent serum? Present references.

7. Why are some references in numerical format (ex (22) (23)) and others are in nominal format (ex Damm et al, 2021)?

7. PLOS authors have the option to publish the peer review history of their article (what does this mean?). If published, this will include your full peer review and any attached files.

Reviewer #1: **Yes: **Carlos Correa-Netto

Reviewer #3: No

---

## [Author Response · Author response to Decision Letter 1]

9 Nov 2021

Additional Editor Comments (if provided):

Based on the new round of peer-round some minor concerns have arisen. I recommend a minor revision before acceptance of the manuscript.

Reviewers' comments:

Reviewer #1: The authors rewrote the manuscript and provide relevant references and endeavored to comply with all referee requests. I just emphasize that only Ovophis don't belong to a taxonomic system. This task dont were revide for authors. However. The authors present a study with high quality in methodology, data generation and data analysis, producing significant contribution for toxinology field. Thus, this work is suitable for publication on PLOS ONE.

Response: We corrected Ovophis to Ovophis spp. as suggested by the reviewer (e.g. lines 68, 83, 91, 288, 305, 313, 396) 

Reviewer #3: The article highlights the spectrometry-based proteomics and the immunoreactivity of the venom of the Ovophis monticola viper. It is a work based on an excellent premise and the methodologies are very well defined. I recommend the work for publication. However, some minor revisions are needed.

1. In the session "Two-dimensional polyacrylamide gel electrophoresis (2DE)" (Material and Methods), the authors state that a 100 µg sample of venom was used. However, since the protein concentrations were measured using the Lowry method, please review whether these 100 µg are venom or protein.

Response: : It was 100 �g of protein. We corrected the sentence in line 135.

2. In the session “Proteomic analysis of O. monticola venom” (Results), the first paragraph deals with a methodological aspect.

Response: We rewrote this paragraph according to the reviewer’s suggestion.

3. The same observation applies to the following topics. Instead of repeating the concepts of the methodology, the objective could be presented if evaluating that result, since this information is not present in the text.

Response: We revised result session and eliminated repetition particularly on methodology.

4. In “Immunoreactivity of protein antigens in O. monticola venom to monovalent and polyvalent antivenoms by indirect ELISA”, the information “Since there is no homospecific antivenom to Ovophis venoms currently available, all pit viper envenoming victims usually receive either monovalent antivenom (raised against T. albolabris venom) or hematotoxic polyvalent antivenom (produced against venoms of C. rhodostoma, D. siamensis and T. albolabris) to alleviate symptoms” is applicable to the discussion, requiring references.

Response: We clarified this point by providing a reference (line 232, ref no. 16 )

5. Figures and tables should be self-explanatory. In the case of table 2, a legend stating the meaning of MW, PI, emPI must be provided. 

Response: We added the explanation in the table 2 legend (lines 238-241, 244-246) and the definition of abbreviation is provided as in a legend as suggested.

6. Discussion: What is the toxinological consequence related to the fact that the venom has more acidic than basic protein spots? How does this interfere with symptoms related to accident? Is this positive or negative regarding the immunoreactivity of the venom, especially with the polyvalent serum? Present references.

Response: In this revised version of manuscript, we discussed the above points raised by the reviewer with relevant references, lines 323-327, 391-395. In addition, to make table 4 more readable, we added a column of MW of non-immunoreactive peptides.

7. Why are some references in numerical format (ex (22) (23)) and others are in nominal format (ex Damm et al, 2021)?

Response: We corrected all references appeared in text in numerical format according to Plos guideline.

---

## [Editor Report · Decision Letter 2]

11 Nov 2021

Proteomics and immunocharacterization of Asian Mountain pit viper (Ovophis  monticola) venom

PONE-D-21-18686R2

Dear Dr. Kumkate,

We’re pleased to inform you that your manuscript has been judged scientifically suitable for publication and will be formally accepted for publication once it meets all outstanding technical requirements.

Kind regards,

Rafael Ximenes

Academic Editor

PLOS ONE
---

## [Editor Report · Acceptance letter]

19 Nov 2021

PONE-D-21-18686R2 

Proteomics and immunocharacterization of Asian Mountain pit viper *(Ovophis  monticola)* venom 

Dear Dr. Kumkate:

I'm pleased to inform you that your manuscript has been deemed suitable for publication in PLOS ONE. Congratulations! Your manuscript is now with our production department. 

Kind regards, 

on behalf of

Dr. Rafael Ximenes 

Academic Editor

PLOS ONE